# First Principles Calculation of Adsorption of Water on MgO (100) Plane

**DOI:** 10.3390/ma16052100

**Published:** 2023-03-05

**Authors:** Bin Li, Hongqiang Chen, Jisheng Feng, Qiao Ma, Junhong Chen, Bo Ren, Shu Yin, Peng Jiang

**Affiliations:** 1School of Materials Science and Engineering, University of Science and Technology Beijing, Beijing 100083, China; 2Zibo City Luzhong Refractory Co., Ltd., Zibo 255000, China; 3Institute of Multidisciplinary Research for Advanced Materials (IMRAM), Tohoku University, 2-1-1 Katahira, Aoba-ku, Sendai 980-8577, Japan; 4Advanced Institute for Materials Research (WPI-AIMR), Tohoku University, 2-1-1 Katahira, Aoba-ku, Sendai 980-8577, Japan

**Keywords:** MgO, surface structure, adsorption, first-principles calculations

## Abstract

The hydration reaction seriously affects the quality and performance of MgO-based products. The final analysis showed that the problem is the surface hydration of MgO. By studying the adsorption and reaction of water molecules on the surface of MgO, we can understand the nature of the problem from the root cause. In this paper, first-principles calculations are performed on the crystal plane of MgO (100) to study the influence of the different orientation, sites and coverage of water molecules on the surface adsorption. The results show that the adsorption sites and orientations of monomolecular water has no effect on the adsorption energy and adsorption configuration. The adsorption of monomolecular water is unstable, with almost no charge transfer, belonging to the physical adsorption, which implies that the adsorption of monomolecular water on MgO (100) plane will not lead to the dissociation of water molecule. When the coverage of water molecules exceeds 1, water molecules will dissociate, and the population value between Mg and O_s_-H will increase, leading to the formation of ionic bond. The density of states of O p orbital electrons changes greatly, which plays an important role in surface dissociation and stabilization.

## 1. Introduction

In the process of converter smelting, the iron side of the furnace lining is frequently impacted by molten iron, resulting in serious corrosion and imbalance of converter refractory. In achieving connection of steelmaking process and improvement the service life of converter, the corrosion area must be repaired quickly with hot repair refractory [1,2,3]. At present, the main contents of hot repair refractory are magnesia and asphalt (asphalt content is about 10~20 wt%) at home and abroad [4]. Due to the excellent fluidity and high adhesion strength at high temperatures of asphalt, this hot repair refractory exhibits excellent performance. However, there are serious problems in the use of asphalt hot repair refractory. Due to incomplete oxidation of asphalt, thick smoke billows during the hot repair process of converter, causing huge pollution to the workplace and surrounding environment. Therefore, researchers have been looking for hot repair refractory of pollution-free as substitutes for magnesia-asphalt refractory.

MgO-SiO_2_-H_2_O is a new inorganic gel material with the advantages of better corrosion resistance, stable at high temperature, good surface gloss, light weight, outstanding mechanical properties and zero pollution; therefore, it has many applications, such as wall insulation materials, packaging of nuclear and metal waste and immobilization of pollutants [5,6,7,8]. The magnesium silicate hydrate gel (M-S-H gel) in MgO-SiO_2_-H_2_O system give excellent fluidity and rheological properties [8,9,10]; at the same time, it also has better high temperature strength, so it has become the important substitute for magnesia-asphalt refractory. Although the MgO system has great application potential in hot repair refractory, there is a fatal problem in this system, which is the hydration of MgO, resulting in very unstable performance. The hydration of MgO prevents the formation of M-S-H gel, resulting in Mg(OH)_2_ precipitation, further deteriorating fluidity and rheological properties of MgO-SiO_2_-H_2_O system [10,11]. Especially in the high temperature weather, the hydration of magnesium oxide is very serious, and the hot repair refractory cannot be used very often. This is not the only problem caused by the hydration of MgO; MgO is an important industrial raw material, and every year, there is serious waste due to the hydration of MgO. More seriously, as an important refractory for steelmaking, the hydration of MgO leads to many accidents, such as splashing and explosion of molten steel.

To clarify the hydration mechanism of MgO, there have been many studies, and most of them focus on the effects of different magnesia [12,13] and various additives [13,14,15] on the hydration reaction under experimental condition [16,17]. At present, there are two main understandings on the hydration mechanism of MgO. The first is the shrinking core mechanism, and the second is the dissolution and precipitation mechanism. According to the shrinking core mechanism [18], Mg(OH)_2_ is quickly generated at the grain boundary as soon as the MgO surface contacts with water. With the gradual increase in Mg(OH)_2_, the pores and free space at the boundary are gradually filled. The volume expansion of MgO transforming to Mg(OH)_2_ causes the internal tension, which become larger and larger. Eventually, MgO will undergo stress fracture and split into finer MgO particles, and the hydration rate will increase rapidly, which is a kind of macro mechanism [19,20]. As for the dissolution and precipitation mechanism [15,21]: the O on the MgO surface attracts H^+^ dissociated in water, forming a proton film on the MgO surface, making the surface positively charged, as shown in Formula (1); the free OH^−^ in the solution rapidly increases the pH value at the initial stage of hydration, and the protonated MgO surface begin to absorb and combine with the OH^−^ (Formula (2)); then, Mg^2+^ and OH^−^ begin to ionize near the MgO surface (Formula (3)). When Mg^2+^ and OH^−^ reach supersaturation, precipitation begins appear on the MgO surface, as shown in Formula (4).
(1)$${\mathrm{MgO}}_{\left(\mathrm{Solid}\right)}{+H}_{2}O\;\to {\;\mathrm{MgOH}}_{\left(\mathrm{Surface}\right)}{}^{+}{+\mathrm{OH}}_{\left(\mathrm{Aqueous}\right)}{}^{-}$$
(2)$${\mathrm{MgOH}}_{\left(\mathrm{Surface}\right)}{}^{+}{+\mathrm{OH}}_{\left(\mathrm{Aqueous}\right)}{}^{-}\to {\;\mathrm{MgOH}}^{+}\cdot {\mathrm{OH}}_{\left(\mathrm{Surface}\right)}{}^{-}$$
(3)$${\mathrm{MgOH}}^{+}\cdot {\mathrm{OH}}_{\left(\mathrm{Surface}\right)}{}^{-}\to {\;\mathrm{Mg}}^{2+}{+2\mathrm{OH}}_{\left(\mathrm{Aqueous}\right)}{}^{-}$$
(4)$${\mathrm{Mg}}^{2+}{+2\mathrm{OH}}_{\left(\mathrm{Aqueous}\right)}{}^{-}\to \;\mathrm{Mg}{\left(\mathrm{OH}\right)}_{2(\mathrm{Solid})}$$

In this process, it is also unknown how the O on the surface of MgO adsorbs and reacts with H_2_O molecule to form bonds step by step, how the protonated MgO surface adsorbs and bonds with OH^−^, and how the Mg^2+^ and OH^−^ are released through structural transformation, which is crucial for the control of hydration reaction on the surface of MgO. In this paper, the interaction between H_2_O molecule and MgO surface is analyzed based on the first principle. Various adsorption possibilities of H_2_O molecule with different distribution forms on MgO surface are fully considered and the adsorption and bond effects of H_2_O/MgO interface were analyzed. In our calculation, the ideal (100) plane was used to study flat surface, and do not involve hydration analysis of surface defects. In fact, not just the flat surface, but a large of the step surface with an atomic height was also common on the MgO (100) plane. At same time, some concave point defects, vacancy and hydrogen-related defects may also exist, which are stable on the flat surface [22,23]. This work does not cover these situations, and only carries out first-principles analysis of the hydration process of the relatively basic ideal surface. The purpose is to propose a new perspective for the hydration reaction of MgO surface under different conditions, and provide a theoretical basis for the research of MgO waterproofing.

## 2. Methods and Models

### 2.1. DFT Calculation Method and Parameters

The method of density functional theory (DFT) is adopted, the plane wave pseudopotential method is used in CASTEP program [24], which is included in Materials studio 6.0, and the generalized gradient approximation (GGA) and the exchange correlation function of Perdew Burke Ernzerhof (PBE) [25] are also used for calculation. For the calculation of H_2_O/MgO interface, the ultrasoft pseudopotential [26] is usually used to describe the interaction between ion nuclei and valence electrons. It can reduce the plane wave number required for Kohn Sham orbital expansion, and at the same time bring about a relative reduction in accuracy. Therefore, this paper chooses the dynamically generated ultrasoft pseudopotential with higher accuracy to calculate, the modified ultrasoft pseudopotential O atom has 6 electrons and Mg atom has 10 electrons. Grimme [27] of DFT-D2 was added to correct the deficiency of van der Waals interaction description in PBE, which can accurately characterize the interaction between surface and water molecules.

### 2.2. Calculation of Adhesion Energy and Selection of Crystal Face

To analyze the growth of MgO crystal more accurately, Morphology module in Materials Studio 6.0 software (version number: v 6.0.0) was used to simulate the crystal morphology and crystal surface of MgO based on dynamic growth [28,29,30]. In crystal growth, the adhesion energy, which is the energy released by the growth unit attached to the growth surface, is often used to indicate the growth speed of the crystal surface. The calculation formula is as follows:(5)$$E_{\mathrm{adhesion}}{=E}_{\mathrm{lattice}}-E_{\sec\mathrm{tion}}$$

Among them, E_adhesion_ is the adhesion energy, E_lattice_ is the lattice energy and E_section_ is the energy after cutting a layer of atoms from the lattice. Negative value of E_adhesion_ represent exothermic, and positive value of that represent endothermic. The more negative the value of E_lattice_, the greater the heat release, the greater the energy released when the crystal surface is attached, and the faster the growth. The possible exposed crystal faces and adhesion energies of MgO crystals are shown in Table 1.

It can be seen from Table 1 that the order of adhesion energy of different crystal plane is (100) > (110) > (210) > (310) > (111), and the importance of crystal plane exposure is (100) > (110) > (210) > (310) > (111). The crystal plane (100) has the largest adhesion energy, which indicates that the energy released by the growth on the surface is small. From the perspective of energy, the crystal plane (100) is relatively stable, and the growth is relatively slow. After the crystal growth is completed, the crystal plane (100) can reach much of the surface. Therefore, the crystal plane we studied in this work is crystal plane (100).

### 2.3. Establishment and Optimization of Model

The lattice constant of optimized MgO cell is 4.198 Å, which is very closed with the experimental value of 4.211 Å, and the error is only 0.3%. The MgO surface adopts a layer geometry model, and the two layers are separated by a vacuum region of 15 Å to prevent the interaction of the two periodic surface models. The supercells of five atomic layers of MgO lattice are composed of (2 × 2) unit, and two layers in the bottom are fixed, as shown in Figure 1a. The upper three atomic layers and adsorbed atoms are fully relaxed until all force components decrease or are less than 0.03 eV/Å, and the change in the total energy of self-consistent field convergence does not exceed 1.0 × 10^−6^ eV/atom. The plane wave truncation energy was 650 eV, and a 4 × 4 × 2 Monkhorst-Pack [31] grid was used for the first Brillouin zone integration in the convergence test. The Slab model is symmetric, which does not lead to the generation of dipoles, so there is no electrostatic interaction between the slab and its periodic image along the z-axis [32]. This model does not require dipole correction to offset the potential energy gradient caused by the surface dipole.

After geometric optimization, the bond length of H_2_O is 0.97 Å, and the bond angle is 104.2°, which is very close to the experimental value of 0.99 Å, as shown in Figure 1b. There are four possible adsorption sites on the MgO (100) plane, namely, O site, Mg site, bridge and hollow, respectively. The electron orbital of H atom participating in the calculation is 1s^1^, O atom is 2s^2^2p^4^, Mg atom is 2s^2^2p^6^3s^2^, and the calculation is carried out in the reciprocal space.

### 2.4. Calculation of Adsorption Energy

The possible adsorption position of H_2_O on the surface of MgO are preliminarily determined by calculating and comparing the adsorption energy of different sites in the DFT calculation process. The calculation formula of surface adsorption energy is as follows:(6)$$E_{\mathrm{ads}}{=E}_{H_{2}O/\mathrm{MgO}}-E_{\mathrm{MgO}}-E_{H_{2}O}$$

$E_{H_{2}O/\mathrm{MgO}}$ is the total energy of H_2_O adsorbing on the surface of MgO, $E_{\mathrm{MgO}}$ is the energy of MgO surface, $E_{H_{2}O}$ is the energy of H_2_O molecules and $E_{\mathrm{ads}}$ is the adsorption energy of H_2_O adsorbed on the MgO surface, which is generally negative. The more negative the adsorption energy, the stronger the adsorption.

## 3. Results and Discussion

### 3.1. Adsorption of Monomolecular Water on Crystal Plane (100) of MgO

As shown in Figure 1b, as to MgO (100) plane, there are four possible adsorption sites, namely, atop-O, atop-Mg, bridge and hollow. The orientation of H_2_O molecules at each adsorption site on the surface is also different, namely, H atom up (H up), H atom down (H down), H atom up down (H up down) and H atom parallel (H parallel), as shown in Figure 2. Therefore, due to different initial orientation distribution of water molecules on the MgO (100) plane, a large number of possible adsorption models can be established. The water molecules are placed at a distance of 3 Å from the MgO (100) plane.

According to different adsorption sites on MgO (100) plane and orientations of water molecules, there are a total of 38 adsorption models; the detail configuration s before and after adsorption are shown in Figure 3. The configuration of water molecules at bridge sites after adsorption in different orientations are very similar with only a few deviations. These differences are very likely caused by the deviation of the initial positions of water molecules. From the configuration after absorption of the Hollow site, Mg site and O site, the conclusion is similar to that of the Bridge site, which preliminarily indicates that the initial adsorption orientation of water molecules on the MgO (100) plane does not change its adsorption form and configuration after adsorption. In order to discuss this problem in depth, the detail information of typical models related to adsorption was provided as shown in Table 2.

From Table 2, the changes in bond length, bond angle, adsorption energy between monomolecular water and surface atoms of MgO (100) plane after adsorption were provided. To distinguish the oxygen in crystals and molecules, in this work, O_s_ represents the oxygen atom on the MgO plane, and O_w_ represents the oxygen atom of water molecules. The distance between H atoms in water molecule and O atoms on the MgO plane is expressed in *d*_H-Os_, the minimum atomic distance is 1.72 Å, which is obviously far from the extent of chemical adsorption. The distance between hydrogen and oxygen atoms is mainly distributed at 1.72~2.03 Å, which corresponds to the value of the adsorption energy, indicating that the adsorption is not strong. The distance between the Mg on the MgO plane and the O atoms in the water molecule is expressed in *d*_Mg-Ow_, and the atomic distance fluctuates between 2.24 and 2.27 Å, which is larger than the bond length of Mg-O bond in MgO [33,34]. On the other hand, the bond length of water molecules adsorbed on the MgO plane changes very little, which also implies that the effect of MgO (100) plane on water molecules is weak.

Table 3 gives the Mulliken charge distribution near the adsorption site on MgO (100) plane. It can be seen that the adsorption of single molecule water on the (100) plane is mainly physical adsorption caused by electrostatic attraction, without charge transfer and no obvious chemical bond exists. This is consistent with the result reported by Alvim et al. that monomolecular water will not dissociate on the clean surface [32]. The change in the density of states before and after the adsorption at the O position (H up down) is shown in Figure 4. In terms of the density of states, there is no significant difference in adsorption results compared with the charge transfer analysis. This is because the calculation condition is the single-molecule adsorption on the perfect MgO (100) plane under the vacuum condition and absolute zero. However, in the actual situation, the surface condition is more complicated due to temperature, gas molecules, and imperfect surfaces, which will not be considered in this paper. Under ideal conditions, the force of monomolecular water on MgO (100) plane is limited.

As shown in Table 3 and Figure 4, after water molecules are adsorbed, some electrons obtained by Mg and the population value of Mg-O decreases from 0.26 to 0.19, meaning bond strength decreases slightly. At the same time, after adsorption, O_s_ obtains very few electrons, bond strength decreases, bond length increases and the change in electrons can well explain the reason for low adsorption energy. From Figure 4a,b, the change in state density before and after adsorption is not obvious. The p orbital of the O atom contributes most of the state electrons near the Fermi surface, where there is a nearly vertical peak. As for the Mg atom, most of the state electrons near the Fermi surface are contributed by the s and p orbitals. The resonance peaks of Mg and O appear near the Fermi surface, but they are not strong, indicating that the Mg-O bond is not tightly bound. From Figure 4c, a peak with the contribution of H atom appears at −8 eV, indicating that the interaction between H and O atom is stable on the (100) plane to a certain extent. After adsorption, the peak of O p orbital shifted significantly to the left, the peak value decreased slightly, and the peak width increased subtly. The monomolecular water represents too low a concentration during the adsorption process; therefore, these changes are very small. In Figure 4b, there is a small resonance peak between H atom and O p orbital, which indicates that H s orbital and O p orbital are hybrid, but the interaction is not strong. In the case of monomolecular water adsorption, the four sites (H up, H down, H up down, H parallel) do not have greater adsorption activity. This result is only applicable to the ideal crystal surface without defects. If there are defects, chemical adsorption or chemical bonding may occur at active sites such as defects.

### 3.2. Adsorption of Multimolecular Water on Crystal Plane (100) of MgO

The adsorption of single molecule water on the MgO (100) plane has been preliminarily understood. On this basis, the adsorption and reaction of multi molecules water on the (100) plane will be studied in this section. The coverage is adopted to describe the concentration of water molecules on the MgO (100) plane, which is defined by the ratio of the number of water molecules to the number of units. The MgO (100) plane in the model is composed by (2 × 2) unit, when a single water molecule is adsorbed on its surface, the coverage is 0.25. Models with five coverages of 0.25, 0.5, 0.75, 1 and 1.25 were calculated and compared.

The adsorption energy of multimolecular water on MgO (100) plane are shown in Table 4. The adsorption energy decreases with the increase in the coverage, indicating that the better the adsorption stability is, the closer the water molecules are bound to the (100) plane.

When the coverage is 0.25, physical adsorption occurs. When the coverage is increased to 0.5 and 0.75, it is found that water molecules still do not dissociate. When the coverage rate exceeds 1, it is chemisorption, and water molecules dissociate into OH^−^ and H^+^ adsorbed at Mg and O sites, respectively. When the coverage is 1, only 50% of water molecules are dissociated, as shown in Figure 5a. When the coverage is increased to 1.25, 0.25 of water molecules are desorbed to the second adsorption layer due to supersaturated adsorption, and the rest of water molecules are adsorbed on the first layer. The first adsorption layer is called the inner Helmholtz layer, and the second adsorption layer is called the outer Helmholtz layer. The water molecules in the two adsorption layers are connected by hydrogen bonds, and only one of them is dissociated. In Figure 5a,b, the nearest distance between the H atom in the water molecule and the O atom in the substrate is 1.03 Å and 0.978 Å, respectively, implying that a hydration reaction has occurred. This also shows that it is difficult to react with water on a MgO (100) plane under good section crystallinity when the coverage is low. Only when the coverage rate increases to a certain extent will the hydrolysis reaction occur. This result is consistent with the calculation results of hydration of water molecules with different coverage in the literature [32,35]. At the same time, the calculation of the potential energy of water molecules with different coverage also proves that this result is credible [36]. The surface interaction including electrostatic attraction, van der Waals force and hydrogen bonding has little effect on the stable MgO (100) plane. Only when there is enough water adsorbed on MgO (100) plane does the interaction force of each adsorption site increase rapidly under the action of multi molecule water. Mg and O atoms become unstable under the action of multimolecular water, and further surface reaction occurs, which is a part of the hydration process.

Figure 6 shows the change in the density of states before and after the adsorption with a coverage of 1.25. The peak value of the green line near the Fermi level (Figure 6a) is mainly attributed to the 2p state electron at the O site. Due to the adsorption and reaction of proton, O_s_-H forms on the (100) plane, which is shown as resonance peaks of O-H in Figure 6b,c, and the positions of these resonance peaks are lower than −8 eV. Compared with the previous single molecule adsorption, the energy is significantly reduced, which is consistent with the trend of gradual reduction in adsorption energy, indicating that the (100) plane gradually tends to be stable with the increase in water molecules. Electron transfer is an important reason for the change in adsorption energy [37,38]. After adsorption, the charge is −0.91e for O and 1.07e for Mg, which proves that the binding force between the 5 coordinated O_s_ and the surrounding Mg atoms is reduced. The bond population between Mg^2+^ and OH^−^ is 0.28, indicating that OH^−^ and Mg form stable ionic bonds after the dissociation of water molecules. Due to the charge transfer, the surface balance is broken, and Mg atoms and O_s_ atoms shift to different degrees, generating more dangling bonds, which lead to the activation of Mg and Os atoms. Meanwhile, after adsorption, the O p orbital shifts seriously to the left, as shown in Figure 6c,d. Compared with the adsorption of single molecules on the (100) plane shown in Figure 4c, the offset is huge. Due to the charge transfer of the surface electric field and the redistribution of the electron density, some atoms or ions can also migrate to more stable positions, leading to surface reconstruction. The protonation of O atoms and the adsorption of OH^−^ by Mg make the surface reach a more stable state, which is also a part of the hydration process.

Similarly, calcium oxide is also an important refractory material, which has not been widely used due to its strong hydration. Compared with the hydration process of MgO surface, the hydration process of CaO is more intense. The relevant DFT calculation shows that (100) plane is also the most exposed crystal plane of CaO. The hydration energy remains constant with increasing coverage on the CaO (100) plane. However, when the coverage of the CaO (100) plane is very low, the surface of CaO (001) will be transformed into disordered state and partially dissolved Ca^2+^ ions. The release rate of cations and OH^−^ is faster and more difficult to control [39,40,41].

## 4. Conclusions

In this paper, DFT calculation of monomolecular and multimolecular water adsorption on MgO (001) plane was carried out, and the influence of water molecules with different orientation and coverage on surface adsorption was analyzed. The main conclusions are as follows:
On the MgO (100) plane, the adsorption energy of monomolecular water with different adsorption sites and adsorption orientations demonstrates little change, ranging from −45 to −60 kJ/mol. The adsorption orientation and sites have no obvious influence on the adsorption configuration. The adsorption of monomolecular water on O site, Mg site, bridge and hollow is unstable, with no obvious active sites and almost no charge transfer, belonging to the physical adsorption. The adsorption of monomolecular water on MgO (100) plane will not lead to the dissociation of water molecule;The adsorption energy increases with the increase in coverage of water molecules. When the coverage exceeds 1, the second layer adsorption occurs. When the coverage of water molecules is 1 or higher, water molecules will dissociate, and the population value between Mg and O_s_-H will increase, leading to the formation of ionic bond. O_s_-H forms stable surface hydroxyl groups, and the density of states of O p orbital electrons changes greatly, which plays an important role in surface dissociation and stabilization, also a part of the hydration process.

## Figures and Tables

**Figure 1 materials-16-02100-f001:**
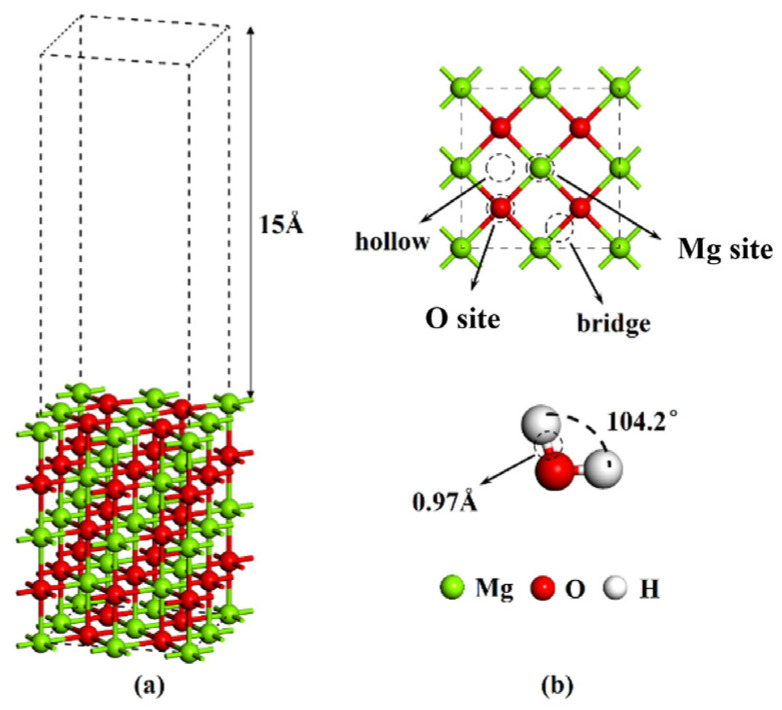
(**a**) Structure model of MgO (100) plane; (**b**) possible adsorption sites of MgO (100) plane and Bond length and bond angle of optimized H_2_O.

**Figure 2 materials-16-02100-f002:**
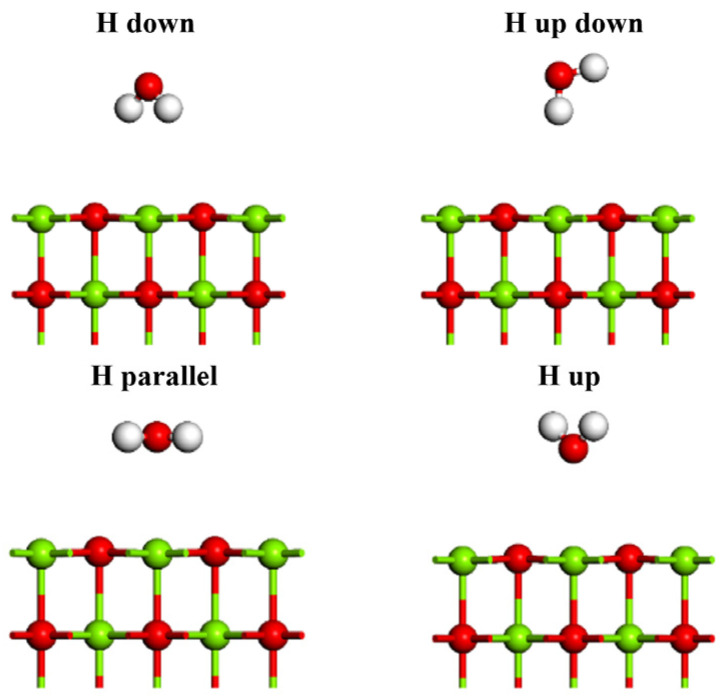
Initial adsorption of water molecules on MgO (100) plane.

**Figure 3 materials-16-02100-f003:**
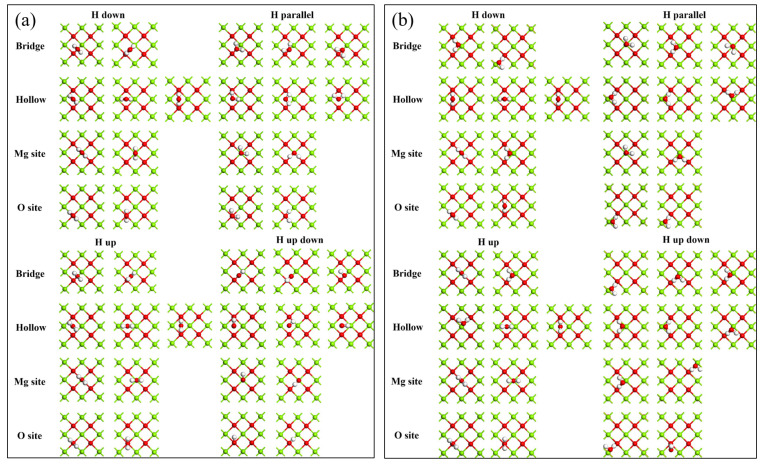
Adsorption models before and after adsorption of monomolecular water on MgO (100) plane: (**a**) before adsorption; (**b**) after adsorption.

**Figure 4 materials-16-02100-f004:**
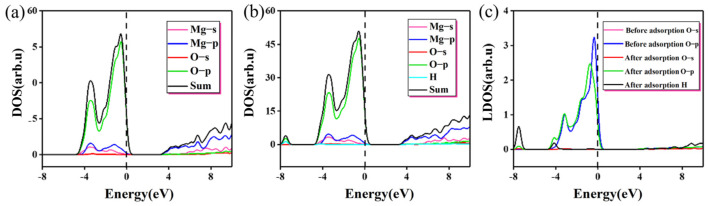
The change in the density of states before and after the adsorption (H up down): (**a**) the density of states before the adsorption; (**b**) the density of states after adsorption; (**c**) change in local density of states before and after adsorption at the O site.

**Figure 5 materials-16-02100-f005:**
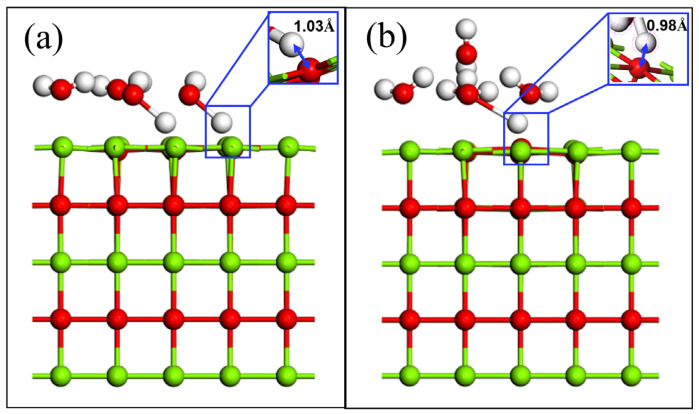
The surface adsorption of multi molecule water: (**a**) coverage is 1; (**b**) coverage is 1.25.

**Figure 6 materials-16-02100-f006:**
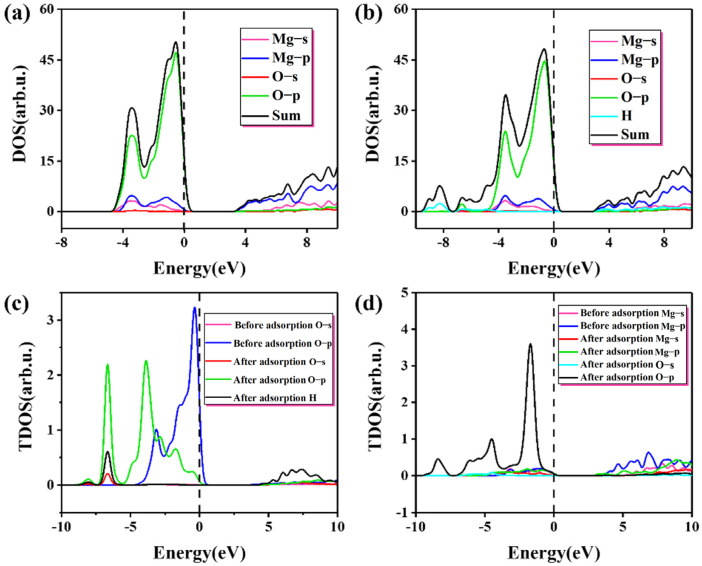
The change in the density of states before and after the adsorption with a coverage of 1.25: (**a**) the density of states before adsorption; (**b**) the density of states after adsorption; (**c**) change in local density of states before and after adsorption at the O site; (**d**) change in local density of states before and after adsorption at the Mg site.

**Table 1 materials-16-02100-t001:** Adhesion energy of different crystal plane of MgO.

Crystal Plane	E_lattice_ (Kcal/mol)	E_section_ (Kcal/mol)	E_adhesion_ (Kcal/mol)
(100)	−3623.48	−3567.54	−55.93
(110)		−3460.72	−162.75
(111)		−1385.16	−2238.31
(210)		−3411.06	−212.41
(310)		−3355.97	−267.50

**Table 2 materials-16-02100-t002:** Structural parameters and adsorption energy of monomolecular water adsorbed on MgO (100) plane with different orientations.

Sites and Orientations	Bond Length on MgO (100) Plane (Å)			H_2_O Bond Length (Å)		*E*_ads_ (kJ/mol)
	*d* _H-Os_	*d* _Mg-Ow_	*d* _Mg-H_	*d* _O-H1_	*d* _O-H2_	
Mg site H parallel	2.0	2.27	-	0.988	0.986	−59.933
O site H parallel	2.0	2.27	-	0.988	0.987	−59.943
O site H parallel	1.77	2.25	2.21	1.005	0.974	−60.627
hollow H parallel	1.75	2.24	2.21	1.007	0.974	−60.616
hollow H parallel	2.03	2.26	2.33	0.987	0.986	−59.868
hollow H parallel	1.79	2.25	2.22	1.003	0.975	−60.586
bridge H parallel	1.80	2.25	2.22	1.002	0.976	−60.557
hollow H up	1.75	2.25	2.21	1.007	0.974	−60.617
bridge H up	1.75	2.25	2.21	1.007	0.974	−60.611
Mg site H up-down	1.74	2.25	-	1.007	0.974	−60.603
Mg site H up-down	1.73	2.24	-	1.008	0.973	−60.562
O site H up-down	1.72	2.25	-	1.010	0.972	−60.503
hollow H up-down	1.94	2.27	-	0.992	0.983	−60.048
hollow H up-down	1.75	2.24	-	1.007	0.974	−60.606
bridge H up-down	1.76	2.24	2.20	1.006	0.974	−60.624
bridge H up-down	1.75	2.24	2.20	1.007	0.974	−60.593
bridge H up-down	1.74	2.24	2.20	1.008	0.974	−60.595
Mg site H down	1.99	2.27	2.32	0.989	0.988	−59.907
bridge H down	1.81	2.25	2.23	1.000	0.977	−60.504
bridge H down	1.74	2.24	2.20	1.008	0.974	−60.588

**Table 3 materials-16-02100-t003:** Mulliken charge distribution near the adsorption site on MgO (100) plane.

Atom	s	p	Total	Charge (e)	Bond Population	Note
O	1.85	5.13	6.98	−0.98	0.26	Before adsorption
Mg	2.38	6.59	8.97	1.03		
O	1.84	5.13	6.97	−0.97	0.19	After adsorption
Mg	2.36	6.57	8.93	1.07		

**Table 4 materials-16-02100-t004:** Adsorption energy of multimolecular water on MgO (100) plane.

Crystal Plane	Coverage	E_ads_ (kJ/mol)
(100)	0.25	−61.632
	0.50	−143.904
	0.75	−200.256
	1.00	−338.784
	1.25	−432.096

## Data Availability

Not applicable.

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
