# Peer review of "First Principles Calculation of Adsorption of Water on MgO (100) Plane"

_materials, 2023, doi:10.3390/ma16052100_

Round 1

Reviewer 1 Report

1. Only 5 articles are 5-years old. New literary sources are needed.

2. How did the authors treat the influence of exchange and correlation effects? Since the standard exchange-correlation functional approximations in DFT do not provide an accurate account of these effects leading to difficulties in accounting for electron localization especially for such strongly correlated systems as the studied ones.

3. Table 3: The parameter "dMg-Ow" in column "Bond length on MgO (100) plane") is almost the same for all the studied cases (within the errors of the simulation technique itself) as well as the angle "H-O-H (deg) ". Could it be the problems of the optimization or may be this table contains no relevant information and may be removed or at least cut?

4.  It is desirable to indicate in Fig. 5 the distances from atoms of H2O to the nearest atom of the substrate.

Reviewer 2 Report

Referee Report on “First Principles Calculation of Adsorption of Water on MgO (100) Plane”

This is, of course, a work that could be recommended for publication, but only after some of the improvements formulated below.

1.       It would be useful to pay more attention in the introduction to vacancy and hydrogen-related defects in the MgO, both in bulk and on the surface. Especially realizing that many other factors act simultaneously with water. See, for example:

González, R., Monge, M. A., Santiuste, J. M., et. al. (1999). Photoconversion of F-type centers in thermochemically reduced MgO single crystals. Physical Review B59(7), 4786.

Ding, Z., & Selloni, A. (2021). Hydration structure of flat and stepped MgO surfaces. The Journal of Chemical Physics154(11), 114708.

Supin, K. K., Saji, A., Chanda, A., & Vasundhara, M. (2022). Effects of calcinations temperatures on structural, optical and magnetic properties of MgO nanoflakes and its photocatalytic applications. Optical Materials132, 112777.

and references therein.

2.     What caused the choice of surface?

3.       It would be useful to compare the obtained results with other calculations, for example, using Crystal or VASP.

4.       There is practically no comparison with experiment in the work, which is a serious drawback.

5.       A fairly large number of hydrogen defects in MgO are known from the literature. Therefore, It is important to see at least short overview.

6.       Comparison with other materials (ZnO, CaO etc) would be essential.
